# Using Theory to Drive Intervention Efficacy: The Role of Dose Form in Interventions for Children with DLD

**DOI:** 10.3390/children9060859

**Published:** 2022-06-09

**Authors:** Pauline Frizelle, Cristina McKean

**Affiliations:** 1Department of Speech and Hearing Sciences, University College Cork, T12 AK54 Cork, Ireland; 2Department of Speech and Language Sciences, School of Education, Communication & Language Sciences, Newcastle University, Newcastle NE1 7RU, UK; cristina.mckean@newcastle.ac.uk

**Keywords:** Developmental Language Disorder (DLD), intervention, theories, dose form, children, vocabulary, morphosyntax

## Abstract

‘Dose form’ is a construct that has evolved over the last number of years and is central to treating childhood language disorders. In this commentary, we present a framework of dose form that includes techniques, procedures, manner of instruction, and intervention context. We present key findings from a systematic review exploring the impact of intervention dose form on oral language outcomes (specifically morphosyntax and vocabulary learning) in children with DLD. We then discuss the hypothesized theoretical mechanisms of action underpinning these findings.

## 1. Introduction

Language interventions for children with Developmental Language Disorder (DLD) are designed to evoke change in a child’s understanding, knowledge, and use of phonology, vocabulary, morphosyntax, or discourse. When designing interventions, the ‘active ingredients’ or ‘dose form’ chosen to evoke such change are influenced by the therapist’s theories regarding the mechanisms underpinning the child’s impairment, the nature of typical language acquisition, and cognitive learning processes. This article highlights key findings from a systematic review published in 2021 on the impact of intervention dose form on oral language outcomes in children with DLD. We present a framework of dose form conceptualisation and summarise key findings from the DLD literature concerning that framework. Additionally, we consider the theoretical underpinnings of or motivations for the dose forms examined in the current literature.

The aim of the systematic review [1] was to extract key points of learning from intervention studies addressing the language difficulties of children with DLD in which the qualitative aspects of dosage (‘dose form’) were manipulated. These are the tasks, activities, and procedures through which intervention teaching episodes are delivered and are referred to as ‘dose form’; the active ingredients hypothesised to bring about change in the child’s knowledge and use of the targeted language goal [2]. The review included studies across the domains of vocabulary, morphosyntax, and phonology, published in any language between January 2006 and May 2020, in which participants with DLD were between 3 and 18 years. Study designs were either quasi-experimental, randomized controlled trials, or cohort analytic. The intention was to examine ‘head-to-head’ studies (i.e., those in which the efficacy of one intervention is compared to an alternative), where dose form was either statistically analysed or experimentally manipulated, while the quantitative aspects of dosage were controlled. In this way, definitive conclusions could be drawn about the relative efficacy of different dose forms, and gaps in the evidence base could be highlighted. However, as we progressed through the process, it was evident that there were no papers in which dose form was manipulated while controlling for all quantitative aspects of dosage. In addition, there was significant variation in how dose form was described and in the levels of detail reported, making it difficult to answer our research questions as definitively as we had intended. However, the process revealed some key points of learning and highlighted aspects of dose form which we deemed to be important but which were not part of any previously published framework. Here we will present the key findings from the review within the context of our framework and will discuss those findings with respect to hypothesised theoretical underpinnings while observing commonalities and differences across domains. We posit that examining the theoretical underpinnings of dose forms that have been tested in the literature has several benefits. It can (1) identify promising directions for future novel intervention development; (2) contribute to hypotheses regarding the underlying nature of the impairment in DLD; and (3) enable clinicians to use this knowledge of theory to ensure they deliver interventions in a manner which can bring about change even when individualising their approaches to address children’s specific profiles.

Note: While the review included vocabulary, morphosyntax, and phonology domains, only one study (with a small sample) reported on dose form manipulation regarding phonological outcomes. We could not draw any firm conclusions based on one study and therefore have not included phonology in this commentary.

## 2. Conceptualising Dose Form

To the best of our knowledge, the concept of dose form was first applied to the field of speech and language therapy/pathology by Warren and colleagues [2], who defined it as the typical task or activity within which the active ingredients believed to effect change are delivered. Proctor-Williams [3] built on this work in expanding the description of dose form to include several different components, namely the techniques, procedures, and intervention contexts that constitute these active ingredients. In our review, we further specified this definition to include other active ingredients which we deemed to be missing from Proctor-William’s taxonomy. We included an additional component addressing the manner (implicit/explicit) in which techniques are delivered, referred to as *Method of Instruction.* We also extended the *Intervention context* component to include the activity in which the technique/teaching episode is being delivered and the degree of variability in the linguistic input or materials used. Our framework and associated definitions are outlined in Table 1. For a more detailed explanation of each component, we refer the reader to our systematic review [1].

## 3. Key Findings and the Hypothesised Theories Underpinning Them

A summary of the findings from each included study is given in Table 2. We will apply the taxonomy outlined in our discussion of these findings and draw out the theories of impairment, language acquisition, and cognitive learning mechanisms, which are either explicitly or implicitly invoked by the dose forms investigated in each paper. While most authors referred to a theory, they varied considerably in the degree to which they provided explicit theoretically motivated hypotheses regarding the mechanism of action of the dose forms examined. Where theory was not explicitly referred to, for example, when comparing one technique combination with another, we reviewed other literature where mechanisms of action were hypothesised for those techniques and presented the main theories subscribed to in the literature.

## 4. Techniques

### 4.1. Vocabulary

Regarding vocabulary, teaching with a phonological versus a semantic focus has been compared [4], as well as teaching with different levels of semantic support (e.g., using dictionary support versus giving explanations in the context of the story). However, the relative effects of these techniques haven’t been established and appear to be dependent on how learning was assessed as well as the language level of the child.

The use of gestural supports appears to be beneficial in the short term [10]. A learning advantage for iconic versus attention-getting gestures has been reported across comprehension, naming, and word definition outcomes [12,13]. However, we do not know if these effects are maintained in the long term. One key theory supporting the use of gestural supports in vocabulary learning is the Dual Coding theory (referred to briefly in [10]). Dual coding theory states that processing linguistic and visual information occurs through the combined action of specialised non-verbal and verbal mental systems and that these systems operate through a rich network of modality-specific representations. In addition, the theory supports the idea that simultaneous input from more than one modality creates a stronger memory trace and therefore facilitates easier access to vocabulary items in the mental lexicon [30]. While pointing gestures focus children’s attention on the referent form they are learning, iconic gestures represent the referent by providing additional semantic information. Given the weak semantic and phonological representations [31,32] and verbal working memory difficulties evinced by children with DLD [33], along with the fact that visual short-term memory and working memory are not always lower than in age-matched peers [34], it is likely that the use of iconic gestures allows children with DLD to capitalise on the facilitative effect of dual coding, to reduce the linguistic load when processing new words. Gestures can therefore serve to scaffold more efficient word learning in DLD.

### 4.2. Morphosyntax

Regarding morphosyntax, studies have compared enhanced conversational recasting vs. recasting, cueing vs. recasting, prompted elicitation with either recasting or modelling vs. recasting alone, and recasting and modelling vs. recasting alone. While each of these techniques offers potential for manipulation to improve efficiency, there is too much variability between studies to unequivocally state that one technique is more effective than another.

Eidsvåg and colleagues [5] compared modelling (where children heard their partner’s target morpheme) to enhanced conversational recast treatment (the technique used to target a child’s own morpheme) and found no gains for morphemes that were modelled only. Proctor-Williams and Fey [11] found no additional benefit to modelling (concerning verb production accuracy) in a procedure that involved modelling + recasting versus recasting alone. The hypothesis as to why recasting is predicted to be effective is based on the idea that recasts are contingent on, and therefore share meaning with, the child’s ‘platform’ utterance. In addition, they share a referential context and involve reformulating components of that utterance. Lastly, they provide an immediate contrast between the child and adult form (on the basis that the recast is corrective) [35,36], thereby minimising working memory demands and facilitating the process of syntactic abstraction. On the contrary, it could be argued that for recasting to be effective, children require a level of metalinguistic awareness to allow them to compare the child and adult forms [14]. In addition, to understand the nature of the correction, the child would need to be focused on the grammatical aspects of the structure.

Smith-Locke et al. [14] found recasting + cueing (which involved eliciting a correct production) to be more effective in the short term than recasting alone. Indeed, regardless of the comparison of techniques used in treatment, whether children get an opportunity to produce the target appears to be key in improving outcomes. While the mechanism by which children are required to produce targets varies between and often within studies (e.g., elicited imitation, prompted elicitation following a recast), the underlying principle is that practice in production is likely to strengthen and stabilize syntactic representations. By eliciting specific ‘practice’ responses, children increase the frequency with which they produce target structures, compared to their production rates in more natural communicative environments. Producing these target forms in a condensed way also highlights their saliency as it is assumed that children with DLD may not readily perceive them in everyday interactions.

## 5. Procedure: Order and Combination of Techniques

Procedure or order effects have not been systematically examined regarding vocabulary interventions for children with DLD. Some effects of procedure manipulation seem to affect morphosyntactic outcomes, which we summarise in the following; however, they remain relatively under-researched.

### Morphosyntax

Initial findings suggest that the order of techniques affects morphosyntactic treatment outcomes. Van Horne and colleagues [16,17] found that beginning treatment with harder to inflect verbs (less frequent, more phonologically complex, and those that describe an action that is not complete, e.g., dropped versus rolled) increases the accuracy of past tense production on both treated and generalisation verbs compared to beginning treatment with verbs that were easier to inflect. Using this kind of ‘complexity-based’ approach is a considerable move away from the more traditional developmental model often employed in practice.

The work is informed by a theory of grammar acquisition that promotes the importance of meaning in influencing the nature of grammatical learning. The authors refer to Goldberg’s early work [37]. She argues that a syntactic frame (such as a ditransitive construction describing events where there is the intended transfer of something to someone) carries a meaning independent of the lexical items in that frame. Building on this principle, the authors also refer to the Frequency Item Template (FIT) hypothesis put forward by Ambridge, Pine, and Rowland [38]. In this hypothesis, Ambridge and colleagues assert that words produced in a grammatical frame are determined by the frequency of the word itself, the frequency of the word within the syntactic frame, and how semantically aligned the word is with the syntactic frame. The suggestion is that children’s acquisition of syntactic frames and morphology is guided by the semantic features of the words being used and the probability of exposure to those words in predictable (well-aligned) frames or constructions [39,40]. In the context of morphological learning, this implies that both the frequency of the verb and the frequency of the verb-morpheme pair are relevant, with those that are less frequently considered to be in keeping with a complexity-based approach. Additionally, the greater the frequency of the verb-morpheme pair, the more aligned it is considered to be (and therefore ‘less complex’). However, although the literature shows that well-aligned verb-morpheme pairings promote the production of early accurate morphological markers in English-speaking children, it is only when children hear morphemes that are less well aligned that they begin to differentiate the morphemes from the verbs that they typically associate with, and so create more abstract and flexible representations of the relevant morphosyntactic structures [41]. Through presenting poorly aligned verb-morpheme pairings, the boundaries of how morphemes are used are more salient, and this is thought to cause a re-organisation of what the morphemes contribute to verb meaning. This theory (and indeed the findings from van Horne and colleagues [16,17]) suggests some dissonance between the linguistic input that supports initial morphological accuracy versus that which results in knowledge of a grammatical rule. Interestingly, Li and Shari [41] argue that the increase in verb vocabulary is the impetus behind the cognitive reorganization that leads to broader morphological accuracy. This line of thought is in keeping with the argument that input variability positively impacts learning [24] and is discussed in more detail in the intervention context section. Because a ‘complexity first’ approach (highlighting meaning contrasts through misaligned verb-morpheme pairings) inadvertently results in enhanced input variability, we cannot be entirely sure what is driving the intervention effects, and it may be that both approaches (complexity and variability manipulation) are harnessing similar underpinning mechanisms to evoke change.

In any case, from a clinical perspective Van Horne and colleagues highlight that although complexity-first approaches to therapy build on principled contrasts, with the aim of achieving generalisation, they are not recommending that a clinician work outside a child’s zone of proximal development or that targets are chosen randomly, such that they are particularly rare. In addition, they highlight the practical difficulty of finding ways to elicit demanding verbs. They conclude with the suggestion that if variability and complexity-driven approaches do rely on the same underlying principle, perhaps a variability-driven approach would be easier to operationalize in practice.

Further research concerning the order of techniques administered has shown that auditory bombardment (a form of modeling involving concentrated high-density presentations of target morphemes in short sentences) is more beneficial after enhanced conversational recast treatment than before [18]. In addition, it was found to be more effective as a therapeutic procedure than recasting on its own. The hypothesis underlying this finding is that modeling post-recasting results in enhanced encoding and stronger consolidation of the targeted morphological information. The authors refer to work by Dudai [42], who suggests that the presentation of intervening stimuli reduces accurate recall of a new targeted form, as it serves to erode an already fragile representation in the child’s memory. Dudai posits that as new memories are being formed, avoiding intervening stimuli as much as possible will reduce interference, but that after a period, the memories could be strengthened by reactivation stimuli. The rapid presentation of models (auditory bombardment) post recasting, without intervening material, could function as reactivation stimuli and strengthen the consolidation of new forms. The benefits of reactivation through repeated recall have been reported in previous studies in the context of word learning, e.g., [43,44]. While it could be argued that modeling does not require children’s active recall of a target form, Plante and colleagues point out that it may result in passive reactivation of related forms.

The idea that intervening stimuli are detrimental to the accurate recall of new targeted forms is specific to morphosyntax and contrasts with that reported in the word learning literature, where intervening material is tightly controlled; presented for study over a set period; and serves to enhance new lexical representations (discussed further under *Intervention context*). Concerning morphological learning, the process is more complex. Following initial targeted exposure, the encoding of how the morpheme is contributing to verb meaning may be so weak that the intervening material serves to further erode the representation. Using a recasting technique, each recast is contingent on the child’s utterance. Therefore, the intervening material is less controlled, and any number of untreated morphemes may be inadvertently presented, each with its own associated rule. We will revisit this discussion under *Intervention context*.

## 6. Method of Instruction: Explicit and Implicit Methods

A comparison of methods of instruction has not been carried out regarding vocabulary for children with DLD, and again we focus on morphosyntax.

### Morphosyntax

Study findings suggest a learning advantage for explicit over implicit instruction given to children with an average age of 7 years. Children who were exposed to techniques administered implicitly but supplemented with explicit instruction, learned not only to use new target morphemes across a greater number of items but did this more quickly and with less intervention than those who were given the intervention with an implicit approach only [19,25].

This finding aligns with the theory regarding the underlying learning deficit in children with DLD. The procedural deficit hypothesis posited by Ullman and Pierpont [45] suggests that many children with DLD have a deficit in their procedural memory. Procedural memory is thought to be integral to the implicit attainment, storage, and use of knowledge. It is hypothesized to be used in implicitly learning the rule-governed features of grammar. In contrast, an additional memory system (declarative memory) is thought to be central to learning explicit information. Studies have shown that while children with DLD have impaired procedural memory, once language and working memory deficits were controlled for, their declarative memory was intact [46]. In addition, while grammatical abilities have been found to correlate with procedural memory in typically developing children, they have correlated with declarative memory in children with DLD. It is therefore suggested that children with DLD may be compensating for their procedural memory deficit by relying more heavily on their intact declarative memory system to facilitate learning the rules of grammar.

The greater effects of explicit instruction also align with theories of DLD as a disorder of linguistic knowledge linked to nativist theory [47,48]. These Linguistic theories would also predict that additional language stimulation only, without using metalinguistic methods (explicit instruction), would have more modest effects on children’s grammatical outcomes. Children included in the studies reported here have an average age of 7 years, and we do not know how the method of instruction interacts with age or treatment progression. It may be the case that younger or more severely impaired children cannot capitalize on explicit instruction as they may not have the metalinguistic awareness to apply the explicit rules presented to them. In addition, the benefits of a given instruction method may change such that implicit methods may be more beneficial during the generalisation phases of treatment.

## 7. Intervention Context: Activity, Child-Centered—Clinician Directed, Variability

### 7.1. Vocabulary

Intervention contexts have been found to interfere with and facilitate children’s word-learning. Giving children with DLD the opportunity to retrieve word names appears to enhance word learning concerning both nouns and adjectives [25,27], and spacing those retrieval opportunities appears to add further benefit [23]. Creating opportunities to retrieve words, as well as spacing word retrievals, are considered ways of altering the context in which words occur, i.e., the intervention context. The theory underpinning these types of intervention is that when children are tested on lexical items and are therefore required to retrieve them, it alters their memory such that it enhances subsequent retention of those items, making it easier for them to retrieve them in the future. Leonard and colleagues subscribe to the episodic context account of retrieval-based learning based on four key assumptions. The first is that words are not encoded in isolation; rather, they include information about the context in which the words occurred [49]. Secondly, when trying to retrieve words, the context, as well as the associated word, are reinstated as part of the memory search process [50]. Thirdly, when the word is retrieved, the representation of the associated context is amended to include features of both the previous and present contexts. And lastly, memory in subsequent retrieval attempts is aided by these updated context representations. The episodic context account also explains why spaced, rather than concentrated retrieval, further benefits word learning [51]. The theory is that (1) the temporal change that has occurred in spaced retrieval requires a greater degree of reinstatement than in concentrated retrieval, and (2) this results in more distinctive context representations [52]. Consequently, how words are represented is less similar to each other, making them easier to access within a reduced memory search space [53]. Work reported here [23,25,27] describes the impact of very slight but tightly controlled non-linguistic contextual changes, which appear to enhance children’s ability to retrieve words. Clinically, it may be easier to operationalise the theory described through less subtle contextual changes, while at the same time being cognisant that too much contextual change may have detrimental effects. In practice, children could be introduced to ‘sets’ of vocabulary in the context of a storybook and be asked to remember and retrieve those words and their attributes/definitions following their first exposure, and again after new words have been introduced.

Concerning the activity within which the techniques are being delivered, video and static stories are equally effective for word learning, but unless presented as a song the presence of music and sounds has been found to interfere with children’s learning [28,29]. Studies have shown that speech perception skills in children with DLD are especially impaired in the presence of background noise (e.g., [54,55], and it is thought that background music and sounds may have similar effects [54]. Although presented as background effects, the music aimed to support children’s understanding of orally presented text by emphasising the mood/emotions within the story. However, children with DLD have shown difficulties identifying basic emotions from music [56]. Therefore the music requires active attention that is not perceived to be linked to the text. In contrast, when presented as a song, the melody and linguistic text are inextricably linked. It is thought that melody can provide an information-rich context that can facilitate the encoding and retrieval of linguistic information [57,58]. This would also be in keeping with the episodic context account previously described. Wallace suggests that not only does a song provide rhythmical information, but it also facilitates chunking text with melodic phrases such that children can associate words or linguistic phrases with specific melodic contours. Because of these associations, the music is thought to aid memory. A familiar melody enables children to recall other contextual information (including linguistic) surrounding their initial exposure to those melodies [59].

Increasing the variability of materials used also seems advantageous in children’s vocabulary learning. Varying how new words are represented (for example, using different objects or pictures to represent the same lexical item) has the potential to improve children’s ability to generalize their word knowledge and to increase the efficacy of the associated intervention [21]. There are two key assumptions underlying this work. Firstly, that children show early perceptual biases that facilitate their word learning [60], and secondly that lexical-semantic mapping is sensitive to statistical information in the input.

Regarding perceptual biases, research suggests that when children learn words and word categories, they recognise multiple perceptual features of a given object (such as shape, function, and material), and these features help to define the semantic class or classes to which the object belongs. However, children with DLD have been shown to have weaknesses concerning the visual perceptions that facilitate word learning [31,32]. Therefore, a treatment approach that increases the saliency of the perceptual features of an object should enhance word learning and improve children’s ability to generalise newly learned labels beyond the specific items that were targeted.

Concerning the second theory—statistical information in the input—there is evidence that input variability can facilitate aspects of word learning [61]. Regarding objects, it is thought that increasing the variability in how an object is presented helps to differentiate between the features peripheral to semantic class membership versus those required. Recognition of the common features associated with each exemplar and its label should reduce the likelihood that children will engage in the one-to-one mapping of words to a specific object form. Rather it should allow for children to form a category that includes many different object exemplars and to generalise new words learned within those categories. By increasing the variability in the objects presented, children can map semantic categories more efficiently, an ability central to learning new words.

### 7.2. Morphosyntax

Variability in the linguistic input is also thought to facilitate grammatical morpheme learning in children with DLD. Plante et al. [24] compared the effects of high (24 unique verbs presented once) and low variability (12 unique verbs presented twice) treatment and found that only those who received the high variability showed a treatment effect. Building on this work, Krzemien et al. [22] found that when learning to generalise constructions, gradually increasing variability in the input (progressive alignment) may be more beneficial for children with DLD than using maximum variability at the outset.

One of the more dominant theories of syntactic learning (usage-based) is that children are sensitive to the linguistic properties of the input they hear and are thought to use this information to extract rules or principles regarding how the input is structured. The supposition is that they progressively learn to generalise concrete linguistic constructions to construct abstract categories/schemas, and they do this through a process of analogical reasoning [62,63]. Analogy is a form of pattern-finding (aligned with statistical learning theories), whereby children identify a common structure between two situations or contexts [64,65]. It also refers to how a new lexical item is slotted within a syntactic frame, allowing for an increase in linguistic productivity [62]. When an analogy is made, the role that the lexical item plays within the dependant structure is more important than the lexical item itself. Children with DLD are impaired in analogical reasoning and, therefore, in generalisation [66]. However, the premise of these interventions is that by modifying the linguistic input for children with DLD, the processes of analogy and generalization can be facilitated. By intently focusing on a small set of linguistic forms and hearing them repeatedly, children can more easily track predictable statistical relations that reflect different grammatical rules. However, within those forms, it appears that repeated presentation of a large variety of unique exemplars (e.g., verbs) is more effective than a small number of exemplars [24,67]. Gómez [67] suggests that with increased lexical variability, children cannot use a learning strategy that relies solely on memory; rather they are induced to transfer their attention to the most stable aspects of the input, i.e., syntactic relations. Therefore, with high lexical variability, the syntactic elements that are frequently repeated become more obvious because of their relative stability. On the contrary, analogical reasoning and generalisation are thought to be facilitated by some degree of semantic similarity [68], and too much variability is thought to impair construction generalisation [69]. By relying on similarity to begin with and gradually increasing the variability of the lexical items presented, the use of progressive alignment is thought to capitalise on these contrary findings. In relation to comprehension, this appears to be borne out in the work by Krzemien and colleagues [22]. Further work is required to explore the impact of this approach on expressive outcomes.

To apply this reasoning clinically, when working on morphosyntactic targets, it may be easier to operationalise by focusing on less common verbs (which inadvertently increases variability compared to what children hear in the ambient language) while ensuring that not all elements of the construction are highly varied, particularly in the early stages of treatment. Given that generalisation is a central goal of clinical practice, the progressive introduction of lexical variability would seem like a fruitful avenue for treating children with DLD.

## 8. Conclusions

To conclude, several common hypothesised mechanisms affect change in morphosyntax and vocabulary domains for children with DLD. With respect to vocabulary, effective interventions discussed here have been designed to create stronger memory traces and enhanced encoding; increase the saliency of perceptual features of an object (through variability); increase the distinctiveness of lexical representations (through context manipulation); and use statistical information in the input to form categories and generalise new word learning. These mechanisms could be adapted and integrated into vocabulary interventions in practice. Regarding morphosyntax interventions, some similar themes emerge. Through increased lexical variability, the saliency of what is stable, i.e., syntactic relations, is highlighted, thereby facilitating the process of analogy and construction generalisation. However, too much variability at the outset is thought to potentially impede this process. In addition, expressive practice is designed to highlight the saliency of target structures (not readily perceived by children with DLD) and strengthen and stabilize syntactic representations through intense repeated target exposure. While this increased saliency is achieved implicitly, children with DLD (with an average age of 7 years) have also been shown to benefit from explicit instructions. Saliency is also a feature of the complexity-based approach, where meaning contrasts are highlighted through misaligned verb-morpheme pairings. Finally, procedures that include techniques administered in a specific order (such as recasting followed by modeling) aim to enhance encoding and strengthen the consolidation of information by reducing interference in the input followed by target reactivation.

Further work is needed not only to identify the most effective dose forms, and whether such effects differ depending on the age of the child or the stage of the intervention. However, subtle manipulations in dose form affect the amount and nature of learning that takes place during an intervention. Many relatively simple practical steps can be drawn from the review and put immediately into practice to increase the effectiveness of interventions, such as varying the referent in vocabulary interventions, providing auditory bombardment of a target morphosyntactic structure after recasting intervention, using explicit teaching at the start of an intervention for children of 7 years or older, and ensuring all interventions offer children opportunities for production of target forms. Furthermore, the development of a sound knowledge of underpinning theory for interventions can support speech and language therapists/ pathologists to deliver these approaches in real-world contexts and with diverse clients in a manner that can effect change. There is also clear potential to draw on theory regarding the mechanisms underpinning DLD, the nature of typical language acquisition, and cognitive processes of learning to design novel interventions to further increase the efficacy and efficiency of interventions and thus improve outcomes for children and young people with DLD.

## Figures and Tables

**Table 1 children-09-00859-t001:** Dose Form Framework and Definitions.

Techniques	The specific teaching behaviours/actions thought to effect change. e.g., providing word definitions (vocabulary), recasting, imitation (morphosyntax).
Procedure	The order or combination of technique delivery. e.g., word exposures followed by word definitions (vocabulary); recasting followed by auditory bombardment (morphosyntax).
Method of Instruction	How techniques are delivered, i.e., implicit only versus implicit plus explicit instructions. e.g., Word exposures alone versus exposures coupled with detailed definitions of targeted words (vocabulary); recasting versus recasting with an explicit explanation of the grammatical rule targeted (morphosyntax).
Intervention Contexts	This has 3 subcomponents The activity within which the teaching behaviour/technique is being delivered, e.g., interactive book reading, play-based activities (both can be adapted for vocabulary or morphosyntax interventions). The location of the activity within a child-centered, clinician-directed continuum. e.g., choosing vocabulary that relates to the child’s interests versus developmentally focused vocabulary; integrating syntactic targets into play-based activities using the child’s toys versus drill-based target games chosen by the clinician. The degree of uniformity or variability in the linguistic input or materials used. e.g., Target vocabulary presented repeatedly with little linguistic variation (many examples of few words) or with greater variability (few examples of many words); manipulating noun and verb variability within syntactic models or recasts provided by the clinician.

**Table 2 children-09-00859-t002:** Summary of ‘head-to-head’ study findings concerning vocabulary and morphosyntax and country in which each study took place.

**Component**	**Vocabulary**	**Findings**	**Morphosyntax**	**Findings**
**Techniques**				
	Korat, O., Graister, T., & Altman, C. (2019) [4] Manipulation of semantic supports for new word learning in e-book activity. Short dictionary explanation vs. explanation given in the story’s context vs. a combination of the two. Country—Israel	Type of semantic support did not affect receptive word learning. Dictionary support most effective for word use in children with SLI. Explanation in context most effective when word definition was outcome. Combined approach best for those with pre-intervention higher language levels.	Eidsvåg, S. S., Plante, E., Oglivie, T., Privette, C., & Mailend, M.L. (2019) [5] Modelling versus Enhanced conversational recast treatment. Treatment given individually and in pairs. Country—United States	Positive effects shown for treatment given individually and in pairs for targeted morphemes. Children in the paired condition showed no significant gains in their ability to produce their partner’s target morpheme (where the target was only modelled). Individual treatment resulted in greater spontaneous use of target morphemes (using enhanced conversational recast treatment).
	Lüke, C., Rohlfing, K., & Stenneken, P. (2011) [6] New word learning in a play context, with prosodic emphasis and semantic elaboration. Using simultaneous sign and speech vs. signs alone. Country—Germany	No statistical difference in number of words learned expressively or receptively 1 week post-intervention, but a statistical trend in favour of the gesture group.	Fey, M. E., Leonard, L. B., Bredin-Oja, S. L., & Deevy, P. (2017) [7] Story modelling, retelling, and recasting with competing input sources (CSI, e.g., competing interrogative form) vs. traditional approach with no competing features. Country—United States	CSI group showed much greater gains in their use of is than traditional group. No significant group differences in the production of 3 s or the control morpheme -ed.
	Steele, S. C., Willoughby, L. M., & Mills, M. T. (2013) [8] word learning task in 4 conditions—phonological vs. semantic vs. phonological-semantic vs. control. Phonological = segmentation and blending tasks (modelled by therapist and then completed by the child). Semantic = child-friendly definitions and use of word associations/synonyms by therapist and child. Combined = phonological and semantic elements as described. Country—United States	Children with LI performesignificantly better in the semantic condition relative to the control condition. Despite the higher dose, phonological condition performance was similar to the control and combined fairly similar to semantic (but not significantly different from control).	Hassink, J. M., & Leonard, L. B. (2010) [9] Variability in conversational recasting—recasts following child utterances that were prompted by clinicians vs. clinicians’ recasts of subject-less sentences vs. clinicians’ noncorrective recasts. Country—United States	Short and long term gains in the use of 3rd person singular were associated with clinicians’ use of non-corrective recasts Recasts of subject-less sentences were associated with poorer outcomes.
	van Berkel-van Hoof, L., Hermans, D., Knoors, H., & Verhoeven, L. (2019) [10] Pseudoword learning with iconic signs vs. no signs, using pre-recording video and pictures. Country—The Netherlands	Children with DLD learned more words with sign than without (immediately post-intervention). Signs did not influence children’s speed of response.	Proctor-Williams, K., & Fey, M. E. (2007) [11] Modelling + recasting v’s modelling alone (in play-based activity). Country—United States	No difference in accuracy of verb production whether recasts were included in the dose form or not.
	Vogt, S. S., & Kauschke, C. (2017a and b) [12,13] Word learning in the context of a story supported by iconic vs. attention-getting gestures (both with speech). Country—Germany	Iconic co-speech gestures improved children’s comprehension, naming, semantic knowledge and word definitions to a greater degree than observing attention-directing gestures.	Smith-Lock, K. M., Leitao, S., Prior, P., & Nickels, L. (2015) [14] Recasting versus recasting + cueing (cueing procedure designed to elicit a correct production following an error to begin with). Techniques differed in the adults’ response to the child’s error). Country—Australia	Cueing + recasting group made significantly more progress than the recasting only group [who showed a negligible effect size] No group differences in maintenance of treatment effects 8 weeks post-treatment.
			Yoder, P. J., Molfese, D., & Gardner, E. (2011) [15] Grammatical recasting (Broad target recasts BTR) versus prompting followed by a recast or model (Milieu language teaching MLT). Country—United States	For children with an MLU of 1.84, MLT was superior to BTR in facilitating grammatical development (despite its lower dose). For children with higher MLU, both treatments yielded similar responses.
**Component**	**Vocabulary**	**Findings**	**Morphosyntax**	**Findings**
**Procedure**				
			Van Horne, A. J. O., Fey, M., & Curran, M. (2017) [16] Complexity-based approach, manipulation in the order of verb presentation, easy to hard, or hard to easy. Country—United States	For overall verb set (target and generalisation verbs), gains in accuracy were significantly greater for hard-first group. Hard first group also made greater gains on all untreated verbs. No differences in time in therapy or progress made on verbs targeted during intervention.
			Owen Van Horne, A. J., Curran, M., Larson, C., & Fey, M. E. (2018) [17] Complexity-based approach, manipulation in the order of verb presentation, easy to hard, or hard to easy. Country—United States	On structured probes, the hard group first advantage (2017) no longer evident at follow-up. Hard group first showed greater gains post-treatment and at follow-up in spontaneous language samples.
			Plante, E., Tucci, A., Nicholas, K., Arizmendi, G. D., & Vance, R. (2018) [18] Enhanced conversational recast treatment preceded or followed by auditory bombardment (high-density presentations of target morphemes in short sentences). Country—United States	More children responded to treatment in bombardment last condition. No significant difference between bombardment. First and last conditions on morpheme use in probes; spontaneous morpheme use; unique utterances containing target morphemes. No generalisation to untreated morphemes.
**Component**	**Vocabulary**	**Findings**	**Morphosyntax**	**Findings**
**Method of Instruction**				
			Finestack, L.H., & Fey, M.H. (2009) [19] Deductive (explicit) v’s inductive (implicit) types of auditory prompts given (using a computer presentation). Country—United States	Although the deductive group heard fewer recasts than the inductive group, more children in the deductive group successfully used the novel morpheme in the teaching probe (10 v’s 3), the generalization probe (10 v’s 3), and the maintenance probe (7 v’s 2).
			Finestack, L. H. (2018) [20] Implicit only vs. implicit + explicit methods of instruction. Country—United States	Explicit instruction enhanced morphological learning. Based on combined performance across 3 targets the explicit-implicit group showed an advantage on acquisition, maintenance, and generalization probes. On individual targets the explicit-implicit group showed a significant advantage on the gender morpheme only.
**Component**	**Vocabulary**	**Findings**	**Morphosyntax**	**Findings**
**Intervention Context**				
	Aguilar, J. M., Plante, E., & Sandoval, M. (2018) [21] Word exposures—through the presentation of physical objects with high variability vs. no variability. Country—United States	Three weeks after the intervention the high variability group was able to identify more objects using new object exemplars of the same class than the no variability group. No differences between groups during the intervention.	Krzemien, M., Seret, E., & Maillart, C. (2020) [22] Exposure to a novel construction (NP- NP- V) in two conditions (high or progressive variability). High variability condition—sentences had no words in common. Progressive alignment condition—initially a proportion of the sentences had words in common, sentences became progressively distinct. Country—Belgium	For the novel construction children with DLD performed better in the progressive alignment condition (at chance) than in the high variability condition (below chance). They also performed better on the transitive than novel construction.
	Haebig, E., Leonard, L. B., Deevy, P., Karpicke, J., Christ, S.L., Usler, E., Kueser, J.B., Souto, S., Krok, W., & Weberb, C. (2019) [23] Word learning using spaced retrieval (practice with contextual changes (RRCR)) versus immediate retrieval without any intervening linguistic material (IR). Retrieval practice involved using picture prompts to recall word names and definition. Country—United States	Despite a lower dose in the spaced retrieval practice condition, word retrieval exercises in which there were intervening words presented, assisted word learning and retention more than repeatedly retrieving and producing a word with no contextual change.	Plante, E., Ogilvie, T., Vance, R., Aguilar, J. M., Dailey, N. S., Meyers, C., Lieser, A.M., & Burton, R. (2014) [24] Manipulation of variability in the linguistic input—high versus low variability in conversational recast treatment. Country—United States	Overall gains were modest. Only those in high variability condition showed significant change in their use of target v’s control morphemes. High variability group also spontaneously produced significantly more inflected verb types. More children in the high variability condition showed a strong treatment effect.
	Leonard, L. B., Karpicke, J., Deevy, P., Weber, C., Christ, S., Haebig, E., Souto, S., Keueser, J.B. & Krok, W. (2019) [25] Word learning using retrieval practice with contextual changes (RRCR) versus repeated study with no retrieval practice (RS). Country—United States	All but one child with DLD recalled more words in RRCR than in RS condition. The RRCR condition also resulted in better word meaning recall. DLD children showed weaker initial coding than TD children but this was no longer evident one week post-intervention.	Riches, N. G., Faragher, B., & Conti-Ramsden, G. (2006) [26] Modelling with either a noun/pronoun or noun-only frame in the subject/object slots (to generalise verbs from a non-transitive to a transitive frame). Country—United Kingdom	Generalisation of the novel verb to a transitive frame was not dependent on the frame used during the training sessions.
	Leonard, L. B., Deevy, P., Karpicke, J. D., Christ, S., Weber, C., Kueser, J. B., & Haebig, E (2019) [27] Word learning using retrieval practice with contextual changes (RRCR) versus repeated study with no retrieval practice (RS). Country—United States	Children with DLD showed higher recall and greater recognition accuracy for adjectives learned in the RRCR condition than in the RS condition—with a large effect. There was no effect of condition for adjective recognition in the TD group.		
	Smeets, D. J. H., van Dijken, M. J., & Bus, A. G. (2012) [28] Experiment 1: Word exposures through electronic stories (static and video, the latter with music and sounds). Experiment 2: Word exposures through electronic stories (static and video, both with and without music and sounds). Country—The Netherlands	Static books were more effective for word learning than those using video with music and sounds (based on a sentence completion task). Video and static stories were equally effective in children’s word-learning. Music and sounds interfered with children’s learning in both contexts. The interference was greater for those with lower levels of language.		
	Kouri, T. A., & Winn, J. (2006) [29] Word exposures through story-telling and acting out. Story scripts given in sung or spoken form. Country—United States	No significant differences in the number of words understood in the sung or spoken conditions (Quick incidental learning). The second sung condition session showed greater spontaneous initiations of novel words.		

Note: More detailed information about each included study is available in the original systematic review [1].

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
