# Peer review of "Using Theory to Drive Intervention Efficacy: The Role of Dose Form in Interventions for Children with DLD"

_children, 2022, doi:10.3390/children9060859_

Round 1

Reviewer 1 Report

The article is a commentary to a review and narrative synthesis on dose form in interventions for children with DLD that the authors have published before. The aim of this commentary is to present a framework of dose form conceptualization (based on their previous review) and to consider the theoretical support for the dose forms examined in the current literature. I see the importance of these goals and appreciate the authors’ overview but have difficulties extracting the “take home messages” of the commentary. In my view the commentary resulted more in an enumeration of different theories that may underpin choices made by other researchers in the discussed interventions. I miss coherence between the individual sections, which may be because the overall framework of dose form conceptualization does not become entirely clear to me. For example, I don’t really understand the difference between the “Procedure” and “Intervention contexts” techniques of the framework. Also, the subparts of the “Intervention context” do feel a bit random to me and very data-driven rather than theory-driven. Furthermore, I think the framework/commentary may benefit from the inclusion of a section that provides a bit more detail on the variability between children that participated in the described intervention studies and how the framework deals with (potential) consequences of this variability (e.g., in ages between children, but most likely also differences in how language impairments are defined across the included/described intervention studies). Finally, I do think the paper may benefit from a more elaborate section on the clinical implications/ take-home messages for clinicians. What interventions have a solid theoretical basis according to the authors and is that something clinicians need to take into account or should they worry more about the efficiency/efficacy of a specific intervention? Other than these general comments, I do have a few specific comments as well:

  • Lines 30-50: it remains a bit vague what is meant with “qualitative aspects of dosage”. It could be helpful to have a few examples (in brackets) of these aspects early on.
  • Lines 109-142: In this section it was a bit unclear to me whether the theoretical embeddings/explanations for the different interventions come from the original papers or that these are theories that the authors of the commentary themselves thought to fit the interventions?
  • Line 154: Is the ‘complexity based’ approach an existing approach/a known approach in the field or is this an approach invented by the authors themselves?
  • Pages 8-9: Though the authors spend quit some words on it, I do not really understand how the FIT hypothesis and complexity first hypothesis are connected.
  • Lines 218-230: I found this section a bit confusing since it starts discussing vocabulary/word learning studies while the section is about morphology.
  • Lines 253-254: In my opinion the conclusion that “children with DLD may be compensating for their procedural memory by relying more on their intact declarative memory system by learning the rules of grammar explicitly” is drawn a bit too fast. Intact declarative memory does not necessarily mean that children learn the rules explicitly (for example because explicit learning requires meta-linguistic knowledge as well. Explicit learning is not equal to declarative learning).
  • I find the last section on intervention context hard to follow because it feels a bit as if all “left-over” information is put into this section. I miss coherence in this section and a clear definition of what intervention context entails. For example, it is not clear to me why the first part about “spacing retrieval opportunities” is discussed in this section rather than in the Procedure section.
  • Line 342: It may be good to highlight that this concerns variability in the non-target structure.
  • Lines 349 – 377: How does this section align with statistical learning theories of language acquisition? The authors speak of analogical reasoning but what they describe sounds very similar to statistical learning.

Author Response

Many thanks for your helpful comments on our manuscript. Please find our responses below.

Reviewer 1. 

The article is a commentary to a review and narrative synthesis on dose form in interventions for children with DLD that the authors have published before. The aim of this commentary is to present a framework of dose form conceptualization (based on their previous review) and to consider the theoretical support for the dose forms examined in the current literature. I see the importance of these goals and appreciate the authors’ overview but have difficulties extracting the “take home messages” of the commentary.

Response: We have added the following text to make these more explicit – ‘We posit that the examination of the theoretical underpinnings of dose forms which have been tested in the literature has a number of benefits. It can 1)  identify promising directions for future novel intervention development; 2) contribute to  hypotheses regarding the underlying nature of the impairment in DLD;  and 3) enable clinicians to use this knowledge of theory to ensure they deliver interventions in a manner which can bring about change even when individualising their approaches to address children’s specific profiles’. See also highlighted text on the final page.

In my view the commentary resulted more in an enumeration of different theories that may underpin choices made by other researchers in the discussed interventions. I miss coherence between the individual sections, which may be because the overall framework of dose form conceptualization does not become entirely clear to me. For example, I don’t really understand the difference between the “Procedure” and “Intervention contexts” techniques of the framework. Also, the subparts of the “Intervention context” do feel a bit random to me and very data-driven rather than theory-driven.

Response: We have now added examples of each component of the framework with respect to morphosyntax and vocabulary to highlight the differences between each construct. We state that Procedure refers to the order or combination of technique delivery, whereas intervention context refers to 1) the activity within which the technique is delivered 2) where the activity is located along a child centred/ clinician directed continuum and 3) the degree of uniformity or variability in the linguistic input or materials used. Two of these sub-components have been previously documented in the literature (see Proctor Williams, 2009; Fey, 1986). We have expanded the concept of intervention context to reflect more recent literature on the effects of linguistic variability (linguistic context) and materials (non-linguistic context) used.

Furthermore, I think the framework/commentary may benefit from the inclusion of a section that provides a bit more detail on the variability between children that participated in the described intervention studies and how the framework deals with (potential) consequences of this variability (e.g., in ages between children, but most likely also differences in how language impairments are defined across the included/described intervention studies).

Response: The aim of the framework is to outline qualitative aspects of language interventions that should be described in order to understand more fully the mechanisms of action in a given intervention; to facilitate faithful replication / a greater degree of implementation fidelity; and to allow for more effective translation into practice. The framework does not attempt to address variability in study participants or how language impairments are defined across studies, rather is about providing a more detailed and consistent description of intervention components. Addressing consistent reporting of participant descriptions would require a different piece of empirical work. Details on the participants / how language impairment was defined is provided in the original review.

Finally, I do think the paper may benefit from a more elaborate section on the clinical implications/ take-home messages for clinicians. What interventions have a solid theoretical basis according to the authors and is that something clinicians need to take into account or should they worry more about the efficiency/efficacy of a specific intervention?

Response: It is not possible to judge from the current research to weigh one theory against another - we posit that a solid understanding of the underlying theory will  enable clinicians to use this knowledge of theory to ensure they deliver interventions in a manner which can bring about change even when individualising their approaches to address children’s specific profiles.

Other than these general comments, I do have a few specific comments as well: Lines 30-50: it remains a bit vague what is meant with “qualitative aspects of dosage”. It could be helpful to have a few examples (in brackets) of these aspects early on.

Response: We have clarified what is meant by the qualitative aspects of dosage earlier in the paper and (see highlighted text page 2). We have added some examples to Table 1 for each component of the framework.

Lines 109-142: In this section it was a bit unclear to me whether the theoretical embeddings/explanations for the different interventions come from the original papers or that these are theories that the authors of the commentary themselves thought to fit the interventions?

Response: We stated on page 5 that ‘While most authors made reference to a theory, they varied considerably in the degree to which they provided explicit theoretically motivated hypotheses regarding the mechanism of action of the dose forms examined. Where theory was not explicitly referred to, for example when comparing one technique combination with another, we reviewed other literature where mechanisms of action were hypothesised for those techniques and presented the main theories subscribed to in the literature’. The papers summarized in this section included some where the authors did not explicitly subscribe to a theory about the technique under manipulation. These papers reported on different interventions in which recasting was one of the techniques used and then compared with another intervention technique (e.g. recasting + cueing vs recasting alone; modelling + recasting vs recasting alone). Where the authors referred to any aspect of an underlying theory we present that and have referenced this to make it more explicit. In addition, we researched theories reported in the literature regarding why recasting might be an effective technique and reference early work carried out by Nelson, which is frequently reported in the literature. We could not present hypothesised theories underlying cueing because of significant variability in how the term is defined i.e. cueing sometimes involves modelling, sometimes imitation, sometimes forced choice questioning etc.

Line 154: Is the ‘complexity based’ approach an existing approach/a known approach in the field or is this an approach invented by the authors themselves?

Response: The concept of a ‘complexity based approach’ has been reported on in a number of previously published papers, with some of the earliest work using complexity based treatment approaches arising from Judith Gierut’s lab in the early 90s. We did not invent this approach.

Pages 8-9: Though the authors spend quit some words on it, I do not really understand how the FIT hypothesis and complexity first hypothesis are connected.

Response: We have added some highlighted text on page 20 to try and make this more explicit.

Lines 218-230: I found this section a bit confusing since it starts discussing vocabulary/word learning studies while the section is about morphology.

Response: We can appreciate how this was confusing and have amended this section (see highlighted text page 22). We refer only to word learning to highlight a contrast -  that while intervening stimuli are detrimental to the accurate recall of new targeted syntactic forms the opposite appears to be the case for word learning, at least under very controlled conditions.

Lines 253-254: In my opinion the conclusion that “children with DLD may be compensating for their procedural memory by relying more on their intact declarative memory system by learning the rules of grammar explicitly” is drawn a bit too fast. Intact declarative memory does not necessarily mean that children learn the rules explicitly (for example because explicit learning requires meta-linguistic knowledge as well. Explicit learning is not equal to declarative learning).

Response: It was not our intention to imply that explicit learning is equal to declarative learning, rather that children with DLD may rely more heavily on declarative memory aspects as a compensatory mechanism. We have rephrased this and it now reads.  …. children with DLD may be compensating for their procedural memory deficit by relying more heavily on their intact declarative memory system to facilitate learning the rules of grammar.

I find the last section on intervention context hard to follow because it feels a bit as if all “left-over” information is put into this section. I miss coherence in this section and a clear definition of what intervention context entails. For example, it is not clear to me why the first part about “spacing retrieval opportunities” is discussed in this section rather than in the Procedure section.

Response: There are three components to intervention context in our Framework and we have now given examples of these in Table 1 which we hope will make it easier to follow this section. Spacing retrieval opportunities are considered to create contextual changes and that is why we discuss them here. We have added the following text to make this more explicit The retrieval of words as well as spacing the opportunity to retrieve,  are considered ways of altering the context in which words occur.

Line 342: It may be good to highlight that this concerns variability in the non-target structure.

Response: This refers to the variability in the verbs and nouns of the target structure. See work by Plante et al., 2014 and Krzemien et al., 2020.

Lines 349 – 377: How does this section align with statistical learning theories of language acquisition? The authors speak of analogical reasoning but what they describe sounds very similar to statistical learning.

Response: We suggest that as a form of pattern finding, analogy is one element of statistical learning. We have now stated explicitly that analogy is aligned with statistical learning theories.

Reviewer 2 Report

Dear authors,

Thank you for the oportunity to read your manuscript. I higly recommend that the method would be conducted towards a review of the literature ou metaanalysis for publication.

Best regards,.

Author Response

Reviewer 2

Thank you for the oportunity to read your manuscript. I higly recommend that the method would be conducted towards a review of the literature ou metaanalysis for publication.

Response: Apologies but we are unsure about what the reviewer means here. We were invited to contribute a paper based on a systematic review that we completed in 2021, on the impact of intervention dose form on oral language outcomes in children with DLD. For the paper presented here we chose to focus on the theoretical underpinnings of / or motivations for the dose forms examined in the current literature. The method for the original paper was a systematic review and narrative synthesis. We refer the reviewer to the original paper for more detail regarding why meta-analysis was not possible or desirable with the available evidence.

Reviewer 3 Report

This manuscript presents a systematic review of the role of dose
form in interventions for children with DLD . Results suggest a lack of consistency in the reviewed studies.
The argumentation is sound, and I did not find anything wrong with the analysis of litterature.

I identified 3 points to improve:
1. In general, I found the paper extremely hard work to read because of many tables without commentaries.

2. The development of pragmatic skills constitutes a major avenue for interventions to better support individuals with DLD but the authors do not develop them, which I think is unfortunate.

3. Linguistic capacities are probably extremely affected by social and cultural aspects. No information is provided on nationalities of individuals with DLD and the authors don’t discuss this issue.

Author Response

Many thanks for taking the time to review our manuscript and for your helpful comments, please find our responses below.

Reviewer 3

This manuscript presents a systematic review of the role of dose form in interventions for children with DLD . Results suggest a lack of consistency in the reviewed studies.The argumentation is sound, and I did not find anything wrong with the analysis of litterature.

I identified 3 points to improve:
1. In general, I found the paper extremely hard work to read because of many tables without commentaries.

Response: We have included two summary tables.

Having downloaded the manuscript in the form prepared for peer review, we note that the tables have been formatted differently to how we presented them. Table 1 was formatted incorrectly, with techniques and its definition presented in bold – as though it was of greater importance than the other components of the framework. This should not have been the case. Similarly, the formatting of table 2 has been altered from our presentation and makes it difficult to read – the component has been centred and it does not make sense, so I can see how this would be confusing to the reader. We have repeated some of the sub-headings to allow for these alterations but would like to discuss this further with the copy editor should the manuscript be accepted for publication.

The first table summarizes our dose form framework (presented in more detail in the publication Frizelle, P., Tolonen, A.-K., Tulip, J., Murphy, C.-A., Saldana, D. & McKean, C. (2021). The Impact of Intervention Dose Form on Oral Language Outcomes for Children With Developmental Language Disorder. Journal of Speech, Language, and Hearing Research, 1–36. https://doi.org/10.1044/2021_jslhr-20-00734). We have now referred the reader to this paper for a more detailed explanation. In addition we have added examples of each intervention component to the table.

The second table summarises the intervention findings of all papers included in the review. Each of these papers and the hypothesised theories underpinning the interventions they describe, are then discussed throughout the remainder of the paper. More detail on the individual studies is provided in the original review cited above. We did not want to repeat information that is available in a different publication and that was not the focus of this work. Again, we have referred the reader to the systematic review for a more detailed explanation.

  1. The development of pragmatic skills constitutes a major avenue for interventions to better support individuals with DLD but the authors do not develop them, which I think is unfortunate.

Response: We agree that pragmatic skills are an important domain of language intervention for children with DLD, however this work was based on our original systematic review which was one of several COST Action IS1406 reviews with differing foci. Pragmatics was the focus of a different review, therefore we did not include it.

  1. Linguistic capacities are probably extremely affected by social and cultural aspects. No information is provided on nationalities of individuals with DLD and the authors don’t discuss this issue.

Response: We agree that linguistic knowledge is affected by social and cultural aspects, for this reason and to capture the dose forms used in interventions that have been published in any language, we included papers from all languages in our original review. The focus of this paper is on the theoretical underpinnings of / or motivations for the dose forms examined in the current literature, and in contrast to many reviews which include only English language papers, our examination of the theories are based on interventions implemented across different cultures. To make this more explicit, we have added the country in which each intervention took place to table 2.